# Identification and Characterization of Peruvian Native Bacterial Strains as Bioremediation of Hg-Polluted Water and Soils Due to Artisanal and Small-Scale Gold Mining in the Secocha Annex, Arequipa

Fernando Fernandez-F [1], Patricia Lopez-C [1], Camilo Febres-Molina [2,3], Pamela L. Gamero-Begazo [3], Badhin Gómez [1,3], Julio Cesar Bernabe-Ortiz [1] and Alberto Cáceres-Huambo [1] and Jorge Alberto Aguilar-Pineda [3,*]

1   Universidad Católica de Santa María, Urb. San José s/n, Umacollo, Arequipa 04013, Peru; ffernandez@ucsm.edu.pe (F.F.-F.); plopez@ucsm.edu.pe (P.L.-C.); bgomez@ucsm.edu.pe (B.G.); jbernabe@ucsm.edu.pe (J.C.B.-O.); acaceresh@ucsm.edu.pe (A.C.-H.)
2   Doctorado en Fisicoquímica Molecular, Facultad de Ciencias Exactas, Universidad Andres Bello, Santiago 8370134, Chile; c.febresmolina@uandresbello.edu
3   Centro de Investigación en Ingeniería Molecular—CIIM, Universidad Católica de Santa María, Urb. San José s/n, Umacollo, Arequipa 04013, Peru; pgamero@ucsm.edu.pe
*   Correspondence: jaguilar@ucsm.edu.pe

**Abstract:** The water and soils pollution due to mercury emissions from mining industries represents a serious environmental problem and continuous risk to human health. Although many strategies have been designed for the recovery or elimination of this metal from environmental sources, microbial bioremediation has proven to be the most effective and environmentally friendly strategy and thus control heavy metal contamination. The main objective of this work, using native bacterial strains obtained from contaminated soils of the Peruvian region of Secocha, was to identify which of these strains would have growth capacity on mercury substrates to evaluate their adsorption behavior and mercury removal capacity. Through a DNA analysis (99.78% similarity) and atomic absorption spectrometry, the Gram-positive bacterium *Zhihengliuella alba* sp. T2.2 was identified as the strain with the highest mercury removal capacity from culture solutions with an initial mercury concentration of 162 mg·L$^{-1}$. The removal capacity reached values close to 39.5% in a period of incubation time of 45 days, with maximum elimination efficiency in the first 48 h. These results are encouraging and show that this native strain may be the key to the bioremediation of water and soils contaminated with mercury.

**Keywords:** mercury removal; bioremediation; native bacterial strains; *Zhihengliuella alba*

## 1. Introduction

Despite being considered one of the most toxic elements for human health [1–3], mercury is still widely used in global artisanal and small-scale gold mining (ASGM) because the extraction procedure for this metal is cheap, simple, and extremely easy to learn [4,5]. In the last decade, ASGM has grown to represent about 12% of the total global gold extraction. This activity is an important livelihood in many developing countries, particularly in regions where economic opportunities for employment are limited [6,7]. However, according to the Pure Earth report, this type of mining activities is responsible for over 30% of the mercury released in the world and one of the main causes of air, soil, and water pollution by heavy metals [8].

One of the stages of gold extraction in ASGM processes is obtaining gold and mercury amalgams. In this stage, the ore with gold concentrations is crushed and mixed with liquid Hg using large amounts of water to wash and settle the processed material. Once the

amalgams are obtained, the water mixed with traces of mercury is discharged into the aquifers or soils. In addition to that, the minimal or lack of knowledge about modern extraction and waste management techniques by artisanal miners increases the risk of mercury poisoning [9,10]. Once used, mercury is often released directly into the environment, where it undergoes a methylation process and is converted into methylmercury, one of the most toxic forms of mercury with devastating ecological consequences [2,10–12]. The impact on human health from mercury contamination is large and well known. Symptoms can range from chest pain or impaired lung function, from exposure to low doses of mercury, to sensory and mental disturbances, neurological and kidney damage, visual or hearing problems, or could even be fatal from exposure to high doses [2,13–15]. Exposure to mercury is especially dangerous in the prenatal stage and in childhood, as it increases the risk of physical deformities, ataxia, coordination problems, neurological damage, and loss of IQ [16,17].

Peru is one of the Latin American countries recognized worldwide for its mining tradition since pre-Columbian times [18]. Lately, despite the decrease in the activities of the global mining industry attributed to the COVID-19 pandemic, Peru ranked as one of the main gold producers worldwide (twelfth, with 88 produced metric tonnes, MT) and fourth in reserves of this precious metal (2700 MT) [19,20]. Gold mining represents almost 1% of the national GDP, the third most important metallic mining activity in Peru. However, the problem of releasing mercury into the environment due to ASGM activities in this country is not an exception, since it is estimated that in 2010 only [21], 70 tons of mercury were dumped into the environment and that the current situation has not changed much.

Among the 25 regions of Peru, three are responsible for more than 70 percent of the official gold production, La Libertad (29.6%), Cajamarca (25.9%), and Arequipa (15%) [20], while the ASGM production represents only 3.2%. However, illegal mining in some Peruvian regions (Puno, Ica, Arequipa, Ayacucho, La Libertad, and Piura) has been increasing, which translates into less control of the waste dumped [22]. Although the ASGM in the Arequipa region is similar to the national average (3.05%), areas such as the Secocha annex have seen an increase in this type of mining activity.

Located in the Mariano Nicolás Valcárcel district in the Camaná province, Arequipa, the Secocha annex has seen a tremendous boost in ASGM activities due to the poor legislation on the use and type of land, which has caused informal settlements dedicated to mining. With high-grade narrow vein gold reserves, where the vein width range is 1 to 10 cm with grades between 10 and 30 g/ton [23], this zone suffers the consequences of gold overexploitation and environmental mercury pollution. In 2018, the residents asked the Ministry of the Environment to declare Secocha in a state of sanitary emergency and requested to identify the impact on the health of the inhabitants due to the high mercury contamination of soils and aquifers [24].

The Peruvian government has implemented various strategies to do so [25–27]. It is worth mentioning that it is one of the countries that signed and ratified the Minamata Convention. In addition, the remediation and sanitation of the different environmental sources are required for the protection of human health [28]. In this context, different techniques are used for this purpose [5,29–31]. However, remediation by microorganisms has proven to be an economical technique, being the most environmentally friendly and a great option for the sustainability of contaminated systems using endogenous bacteria [32–35].

Bioremediation relies on the ability of a few microorganisms to reduce the toxic ionic and organic forms of mercury to less toxic compounds through enzymatic reactions. For example, under certain conditions, some bacteria can secrete exopolymers that adsorb the toxic form $Hg^{2+}$ [35,36]. There are many studies of bacteria that show potential in the removal of Hg from contaminated water, however, in general terms, the bioremediation of soils contaminated by Hg has been little investigated [33,37,38]. Added to this is the use of native bacteria, as some microorganisms develop resistance to Hg contamination, making them promising natural candidates for bioremediation [39–42].

The objective of this study was to identify native bacterial strains from soils contaminated with Hg collected in the Secocha annex and to evaluate their behavior and Hg removal capacity. To achieve our goal, several bacterial strains were isolated and subjected to various processes and analyses to validate our hypothesis.

## 2. Materials and Methods

### 2.1. Study Area

This research was developed in the Secocha annex, located in the Mariano Nicolás Valcárcel district of the Camaná province in Arequipa, Peru (15°58′54.13′′ S, 73°10′21.25′′O) (Figure 1). At an altitude of 348 m above sea level, the Secocha annex borders the district of Ocoña and the provinces of Caravelí and Condesuyos. It is located at a distance of 286 km from the city of Arequipa, and has the towns of Urasqui (capital), Jayhuiche, Cerro Barroso, Secocha, Misky, San Martín, and Venado.

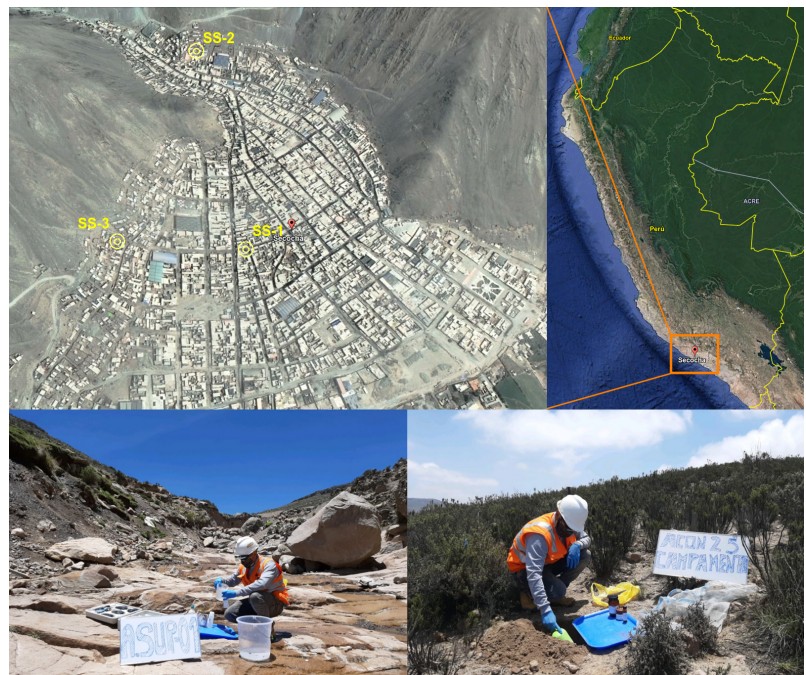

**Figure 1.** The Secocha Annex is located in southwestern Peru, in the province of Camaná. Soil samples were obtained at different sites according to the ARMA report carried out by the Ministry of Environment of Peru (SS-1, SS-2, and SS-3).

In this area, men work as artisanal miners and women as "pallaqueras", a name given to women who manually select the rocks that contain precious metals. In this artisanal mining emporium, there are about 4000 women who work looking for gold among the waste. In addition, there are gold purchasing and/or processing plants for the raw material extracted from the mine pits or from the tailings process that operates there.

### 2.2. Sampling and Isolation of Native Bacterial Strains

The sampling was based on the report "Analysis of soils in points of interest", carried out by the Ministry of the Environment (specifically by the Environmental Regulatory Authority—ARMA) and according to the regulation D.S. N° 011-2017 MINAM-Environmental Quality Standards (ECA) for soils [43]. In this report, the soil quality of 53 samples from different areas of the Secocha annex was measured. The results made it possible to identify the main areas with the highest concentration of metal pollutants. In the case of mercury content for commercial, industrial, or extractive soils, the reported concentrations presented the highest value of 49 mg·kg$^{-1}$ (dry weight), 24 mg·kg$^{-1}$ being

the maximum value allowed for this type of soil, according to the legislation in force in Peru (Table 1).

**Table 1.** Maximum allowed values (ECA) and measured values (MV) for heavy metals according to the ECA of D.S.N°011-2017 MINAM-Soil Quality in Peru.

| Soil Type | As | | Cd | | Hg | | Pb | |
|---|---|---|---|---|---|---|---|---|
| | ECA | MV | ECA | MV | ECA | MV | ECA | MV |
| Agricultural | 50.00 | 75.80 | 1.40 | 2.02 | 6.60 | 12.00 | 70.00 | 76.60 |
| Residential, Park | 50.00 | 75.80 | - | - | 6.60 | 7.70 | 140.00 | 234.00 |
| Commercial, Industrial, Extractive | - | - | - | - | 24.00 | 49.00 | - | - |

All values are in mg·kg$^{-1}$ (dry weight).

With this information, three sites with high Hg contamination were chosen to obtain the soil samples in the Sechocha annex. The first 3 cm of soil was removed and samples were taken down to a depth of about 15 cm. The samples were sieved with a mesh of 1 mm of opening, in order to eliminate unwanted material and they were placed in sterile containers, labeling each one of them. For their study in the laboratory, the containers were stored and transferred in thermal boxes where they were kept at a temperature of 4 °C.

To prepare the suspension and obtain the $HgCl_2$ stock solution, 1 g of $HgCl_2$ was dissolved in 100 mL of sterile deionized distilled water. In addition, two nutrient media were prepared, nutrient agar (4.6 g nutrient Agar powder (Difco$^{TM}$), 200 mL distilled water, pH 6.8), and blood agar (8 g base blood agar (Merck$^{TM}$), 10 mL of sterile defibrinated sheep blood, 200 mL distilled water, pH 7.3). The isolation of bacterial strains was conducted using a serial method by depletion in plates. The three soil samples were mixed in a UV-sterilized bag and 10 g was suspended in 10 mL of sterile deionized distilled water. Dilutions were spread onto nutrient agar containing 100 µL of $HgCl_2$ stock solution and were incubated at 35 °C for 72 h in an aerobic environment. Then, three visually distinct single colonies were streaked 10 times on the same type of agar plates to obtain pure bacterial cultures. A resistant native bacterial strain named T2.2 exhibited a high Hg removal from the culture solution and was selected for the present study.

### 2.3. Purification

By this procedure, a single type of microbial agent can be obtained in a culture in an isolation or culture medium. For this, Petri dishes with nutrient agar were used, to which 5 ppm of $HgCl_2$ was added, the preparation method was the same as that described above. A small amount of the bacterial colony to be purified was taken with an inoculation loop and seeded by exhaustion on a plate, incubated for 72 h at 35 °C. This procedure was repeated 10 times to obtain a pure colony without coexisting microorganisms in the original sample that may vary the molecular sequence. Each of the three isolated strains was subjected to purification following the procedure indicated by Stanchi [44].

### 2.4. DNA Extraction, Sequencing, and Molecular Identification of the Resistant Bacterial T2.2

DNA extraction was performed at the Uchumayo DNA laboratory in Arequipa, Peru. The technique used for extraction is known as the optimal bacterial DNA isolation technique using bead-beating described by Fujimoto et al. [45]. The beads cell disruptor Micro Smash (Tomy Seiko Co., Ltd., Tagara, Nerima-ku, Tokyo) was used to mechanically break cell walls. Glass microspheres were placed in 2 mL sample tubes and a suspension of the bacteria in the buffer was added to the tubes, which were then tightly capped and the samples were beaten at a speed of 4800 rpm for 30 s with 100 mg of 0.5 mm diameter glass beads.

Once the DNA from the bacterial strains was obtained, it was sent to the Molecular Biology Lab at Harvard University, USA, where the 16S ribosomal RNA (rRNA) gene sequencing analysis was performed to identify the Hg-resistant bacterial strain T2.2 to obtain its sequence.

The sequence of the 16S rRNA gene of the bacteria strain T2.2 was sent to the NCBI GenBank database to be aligned and compared with sequences of stored bacterial consortia to determine their degree of homology; the BLASTn algorithm was used for this purpose [46]. Then, a sequence match algorithm was used to determine the regions of similarity between biological sequences, and using the neighbor-joining method, the phylogenetic tree of the bacterial strain was constructed from the calculated routes.

### 2.5. Morphological Characterization and Determination of Antibiotic Sensitivity and Resistance Pattern of Strain T2.2

The morphology of the T2.2 bacterial colony was verified considering the following parameters: color, size, shape, border, and elevation; pH and optimal growth temperature were also verified. In addition, the Gram stain technique, catalase, and oxidase tests were used to identify the isolated strain.

Antibiotic resistance and sensitivity activity tests were performed by the Bauer–Kirby disk diffusion method [47] using Müller–Hinton agar. A quantity of 25 mL of this agar culture medium was introduced into 100 mm diameter Petri dishes, resulting in an agar depth of 4 mm. After inoculation with bacterial colonies, the antibiotic filter paper discs were symmetrically distributed across each other and incubated for 48 h at 37 °C. Eight antibiotic discs were used: ciprofloxacin, lincomycin, azithromycin, amikacin, amoxicillin, levofloxacin, enrofloxacin, and amoxicillin + clavulanic acid. Finally, the strains were classified into three categories: sensitive, medium, and resistant, according to the diameter of the zone of inhibition given in the table of standard antibiotic discs.

### 2.6. In Vitro Evaluation of the Mercury Resistance of Strain T2.2

In vitro experiments were carried out to evaluate the mercury removal capacity of the bacterial strain T2.2 in contaminated soils. To carry out the tests, 200 g of contaminated soil were diluted in sterile distilled water, leaving it to rest for 24 h. The solution was filtered with filter paper with a pore size of 20 μm and the filtrate was sterilized at 121 °C for 20 min. Taking 500 mL of this filtrate, Hg at 162 ppm and 1 mL of the T2.2 strain were added in a Mcfarland turbidity ratio of 1. The containers with the suspension were placed on a horizontal shaker for durations of 12, 24, 48, 720, and 1080 h, in which 15 mL of suspension was extracted to be subjected to ultracentrifugation at 10,000 rpm for 10 min. Then, 10 mL of supernatant was extracted and placed in sterile Falcon tubes to perform the mercury determination at each time lapse using the cold vapor spectrometry method.

### 2.7. Mercury Removal Assays

To determine the Hg removal capacity by the T2.2 strain, a graphical model of the amount of adsorbate on the surface of the adsorbent as a function of the adsorbate concentration in the solution was used. To obtain the necessary data, the measurements were carried out at 12, 24, 48, 720, and 1080 h. To identify the Hg adsorption (capacity and percentage), the following equations and parameters were used:

$$q_t = \frac{V_{(L)}\left(C_i - C_f\right)_{(mg/L)}}{S_{(g)}} \tag{1}$$

$$\%Adsorption = \frac{\left(C_i - C_f\right)}{C_i} \times 100\% \tag{2}$$

where $q_t$, is the adsorption capacity expressed in mg·g$^{-1}$; $V_{(L)}$, is the solution volume expressed in liters; $S_{(g)}$, is the amount of bioadsorbent expressed in grams; $C_i$ and $C_f$ are the initial and equilibrium concentrations of the dye in the solution, respectively, in mg·L$^{-1}$.

### 2.8. Adsorption Kinetics Models

Adsorption kinetics is the fundamental part of biosorption processes, as it defines the rates and mechanisms of heavy metal adsorption by bacterial strains. In the present investigation, two mathematical kinetic models were considered, namely, a pseudo-second-order kinetic model also known as intraparticular diffusion model, based on the theory proposed by Weber and Morris, which is applicable to higher porosity biosorbents. The second model was based on an allometric function using the Levenberg–Marquardt [48] iteration algorithm. Allometric functions are widely used in biological systems related to nonlinear growths [49–51].

The equation used in the Weber–Morris pseudo-second-order model was:

$$q_{W-M} = K_d * t^{0.5} \tag{3}$$

where $K_d$ is the intraparticle diffusion constant ($mg \cdot g^{-1} \cdot time^{-0.5}$). $K_d$ constant values can be calculated from the slope of the linear regions of the graphs $q$ vs. $t^{0.5}$.

The allometric equation was obtained using OriginPro 2020 software, which can perform curve fitting and peak analysis by creating a mathematical model according to the experimental data. In addition, this software program allows the global adjustment of the data with the exchange of parameters and baseline corrections. The equation used was the following:

$$q_A = A * t^B = 6.686 * t^{0.319} \tag{4}$$

where the parameters $A$ and $B$ are values obtained from the iterative fits of the experimental data.

### 2.9. Statistical Analysis

The data obtained from the experimental tests of bacterial growth and their Hg elimination capacity were analyzed using SPSS v.25.0 software by means of a block design for each sampling time [52]. An analysis of variance (ANOVA) was performed to obtain statistical significance and Tukey's tests were used to detect significant differences between means, where obtaining a value of $p < 0.05$ indicated statistical significance. To obtain the kinetic model of adsorption, the experimental data were adjusted to two mathematical models and their coefficients of determination, $R^2$, were evaluated. The models used were a pseudo-second-order kinetic model, according to the Weber–Morris method [53], and an allometric model. For this, OriginPro 2020 software [54] was used, which adjusted the experimental variables to predict the Hg removal behavior of the T2.2 strain at any time during the trials.

## 3. Results and Discussion

### 3.1. Culture, Isolation, and Identification of Bacterial Strains

Three bacterial strains with growth capacity were obtained on nutrient agar and blood agar substrates at 5 ppm Hg concentration. Among them, the strain identified as T2.2 was chosen for further analysis because it showed a higher Hg removal capacity [55,56]. Molecular identification of this strain was performed by 16S rRNA sequencing and the partial DNA sequence (945 base pairs) was analyzed using the BLASTn algorithm. The results in the sequence alignment showed that the isolated strain T2.2 is closely related to the members of the genus *Zhihengliuella*, sharing a sequence similarity of 99.78% with the strains *Zhihengliuella alba* KU147427.1 [57], NR_044575.1 [58], and MK737333.1 [59]. Table 2 shows the top 10 bacterial strains sorted by percent identity; the E-values for all alignments were 0.

**Table 2.** The similarity of the top 10 bacterial strains.

| Description | Total Score | Query Cover | Percent Identity | Accession Length | Accession |
| --- | --- | --- | --- | --- | --- |
| *Z.A.* strain DQ70 | 1705 | 99 | 99.78 | 1409 | KU147427.1 |
| *Z.A.* gene | 1705 | 99 | 99.78 | 1484 | AB778263.1 |
| *Z.A.* strain YIM 90734 | 1705 | 99 | 99.78 | 1454 | NR_044575.1 |
| *Z.A.* strain B4b42 | 1668 | 97 | 99.78 | 1344 | MK737333.1 |
| *Z.A.* strain B1kh77 | 1703 | 100 | 99.68 | 1379 | MK737182.1 |
| *Z.A.* strain J0f78 | 1701 | 100 | 99.68 | 1366 | MK737136.1 |
| *Z.A.* strain TN-Gafsa-I5-P27f-1392r-20091105 | 1701 | 100 | 99.68 | 1289 | GU451719.1 |
| *A.B.* strain 16S10-1 | 1657 | 98 | 99.45 | 921 | MT682454.1 |
| *Z.A.* strain BGR22 | 1589 | 94 | 99.21 | 1312 | KC789781.1 |
| *Z.A.* strain 210-LR22 | 1672 | 99 | 99.14 | 1437 | MF077154.1 |

*Z.A.: Zhihengliuella alba; A.B.: Actinobacteria bacterium.*

The phylogenetic tree generated with the top 10 alignment sequences using the neighbor-joining tree method is shown in Figure 2. Taxonomically, *Zhihengliuella* is a high GC (>60% of the guanine-cytosine ratio) Gram-positive genus of the Micrococcaceae family, consisting of five species, *Z. halotolerans* [60], *Z. alba* [58], *Z. somnathii* [61], *Z. salsuginis* [62], and *Z. flava* [63], an environmental sample (*Zhihengliuella sp.* enrichment culture clone AVCTGRB8A), and 21 unclassified species. Although this genus is frequently associated with plant growth-promoting rhizobacteria (PGPR) and plant defense [64–67], some studies mention its potential capacity to remove heavy metals such as cadmium, lead, zinc, iron, manganese, and other pollutants [68–70]; however, these studies do not include the biodegradation of mercury (or other contaminants) in soils. Therefore, the results show that the T2.2 strain could be a promising candidate for the bioremediation of soil contaminated with Hg.

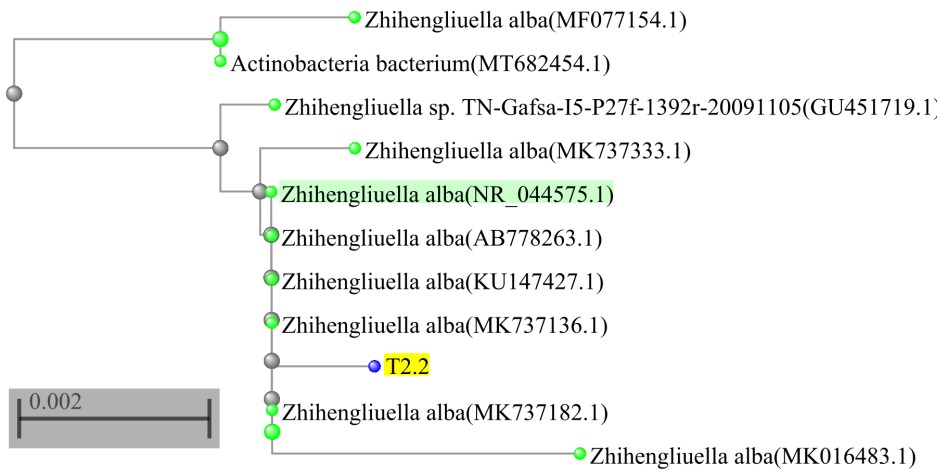

**Figure 2.** Phylogenetic tree based on 16S rRNA gene sequence using the neighbor-joining method showing the position of strain T2.2 and the closely related recognized species.

### 3.2. Morphological Characterization and Antibiograms of T2.2

When evaluating the morphology of the T2.2 strain, it was observed that it has a whitish color with a diameter size of 2.20 mm. Microscopic characterization revealed a circular shape with regular edges and a convex elevation. Biochemical tests showed that this strain was Gram-positive, oxidase-negative, and catalase-positive. Although several studies mentioned that resistance to Hg is common among Gram-negative bacteria, with a lower percentage in Gram-positive bacteria [59,71,72], recent work on volatilization efficiency of Hg showed that its aforementioned resistance does not depend on a specific bacterial group but on specific genera [73].

The optimal growth temperature of the T2.2 strain was 35 °C in a pH range between 8 and 9. This strain was sensitive to five antibiotics (Table 3), showing a zone of maximum inhibition with the antibiotic azithromycin with an inhibition diameter of 38 mm that exceeded the sensitive standard by 20 mm. Levofloxacin showed the second largest diameter of inhibition (36 mm) exceeding the standard by 19 mm, followed by enrofloxacin with a diameter of inhibition of 32 mm, exceeding the standard by 9 mm. Amoxicillin and amoxicillin + clavulanic acid exhibited the lowest zone of inhibition against the T2.2 strain, showing some inhibition effect after 48 h of incubation. The diameter of the zone of inhibition of the antibiotic amikacin was 14 mm, the only one to remain within the intermediate category range (Figure 3). These results are in good agreement with those obtained by Chauhan et al. [74], who showed that bacteria resistant to heavy metals such as Hg are also resistant to antibiotics from the penicillin group (e.g., penicillin, ampicillin) and sensitive to the quinolone group (e.g., levofloxacin, ciprofloxacin).

**Table 3.** Antibiogram of several antibiotics applied to the T2.2 strain.

| Antibiotic | MIC | [a] Inhibition Zone R | S | [a] Measure Diameter | [b] Category |
|---|---|---|---|---|---|
| Ciprofloxacin (CIP) | 5 µg | 15.0 | 21.0 | 28.0 | Sensitive (7) |
| Lincomycin (MY15) | 2 µg | 16.0 | 21.0 | 26.0 | Sensitive (5) |
| Azithromycin (AZM) | 15 µg | 13.0 | 18.0 | 38.0 | Sensitive (20) |
| Amikacin (AN30) | 30 µg | 14.0 | 17.0 | 14.0 | Intermediate |
| Amoxicillin (AML10) | 25 µg | 11.0 | 14.0 | 0.0 | Resistant |
| Levofloxacin (LEV) | 5 µg | 13.0 | 17.0 | 36.0 | Sensitive (19) |
| Enrofloxacin (ENR5) | 5 µg | 16.0 | 23.0 | 32.0 | Sensitive (9) |
| Amoxicillin + Clavulanic ac. (AMC30) | 30 µg | 13.0 | 18.0 | 0.0 | Resistant |

[a] Inhibition zone and measure diameter are in mm. R = resistant, S = sensitive. [b] Values in parentheses represent millimeters that exceed the sensitivity inhibition diameter.

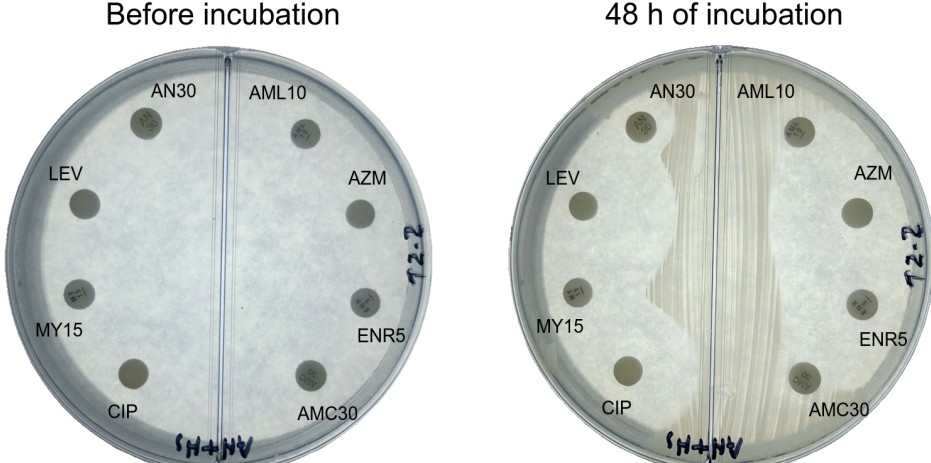

**Figure 3.** Petri dish with the antibiotic discs distributed symmetrically in the medium inoculated with the T2.2 strain. The system was incubated for 48 h at 35 °C.

*3.3. Removal of Hg in Culture Solution by the T2.2 Strain*

To demonstrate the effect of mercury removal by the T2.2 strain isolated from contaminated soils, an in vitro evaluation of the mercury resistance of this bacterial strain was carried out. The test results were obtained from the cold vapor atomic absorption spectrometry test. The experimental results obtained showed a high mercury removal capacity in the first 12 h of the interaction with the T.2.2 strain with respect to the concentration of the stock solution, presenting the highest removal rate (1.25 mg· g$^{-1}$· h$^{-1}$). After 48 h of interaction, a removal percentage of 16.67% (0.5 mg· g$^{-1}$· h$^{-1}$) was reached, with a decreasing trend

from this point on. In the following intervals, an almost linear behavior was observed in the removal of Hg ($\sim$0.04 mg$\cdot$ g$^{-1}\cdot$ h$^{-1}$), reaching a maximum removal of 40% at 1080 h, (Table 4).

This Hg removal capacity of the T2.2 strain may be low compared to two studies carried out using bacterial strains of the genus Pseudomonas, obtained from Peruvian soils [75,76]. These studies showed that the Hg removal capacity reached 90% in soils with a Hg concentration of 100 ppm, whereas for the removal capacity in an aqueous solution at a concentration of 5 ppm, removal values of up to 97% were obtained. However, there are studies that show that high concentrations of Hg inhibit the bioremediation of bacterial strains [75,77], which would explain the lower percentage of removal of Hg from strain T2.2.

**Table 4.** Mercury adsorption capacities (q) and removal rates of the T2.2 strain.

| Time | Experimental Values | | | [a] Weber–Morris Model | | | [b] Allometric Model | | |
|---|---|---|---|---|---|---|---|---|---|
| | $q_{exp}$ | % ads. | Rate | $q_{W-M}$ | % ads. | Rate | $q_A$ | % ads. | Rate |
| 0 | 0.00 | 0.00 | - | 0.00 | 0.00 | - | 0.00 | 0.00 | - |
| 12 | 15.00 | 9.26 | 1.25 | 7.01 | 4.33 | 0.58 | 14.78 | 9.12 | 1.23 |
| 24 | 15.00 | 9.26 | 0.00 | 9.91 | 6.12 | 0.24 | 18.44 | 11.39 | 0.31 |
| 48 | 27.00 | 16.67 | 0.50 | 14.02 | 8.67 | 0.17 | 23.01 | 14.21 | 0.19 |
| 720 | 52.00 | 32.10 | 0.04 | 54.28 | 33.51 | 0.06 | 54.64 | 33.73 | 0.05 |
| 1080 | 64.00 | 39.51 | 0.03 | 66.48 | 41.04 | 0.03 | 62.19 | 38.39 | 0.02 |

[a] The Weber–Morris model values were obtained according to the intraparticular diffusion equation. [b] The allometric values were obtained using Levenberg–Marquardt algorithm.

### 3.4. Kinetics Studies of Hg Removal

The results obtained from both kinetic models at the experimental times are shown in Table 4. Figure 4a shows the adsorption capacities (q) obtained experimentally and from both kinetic models at the different contact times analyzed. In the experimental data (blue line), it can be observed that there is a fast biosorption in the initial 12 h of contact. In the period between 12 and 24 h, practically the same value is registered for the adsorption capacity, which would mean the saturation of the bacteria by slow diffusion. However, adsorption increases again in the next 24 h with a biosorption rate of 0.5 mg$\cdot$ g$^{-1}\cdot$ h$^{-1}$. At this point, the Hg removal capacity becomes slow but constant and remains so until the 1080 h. This trend is due to the fact that, at the beginning of the test, the bacterial strain has a large number of unsaturated sites available for adsorption, which decrease over time. Furthermore, the repulsive forces of adsorbed Hg molecules could inhibit adsorption at other sites [78]. This type of prolonged removal would be very useful in the bioremediation of contaminated soils because it would ensure the efficiency of the treatment by being active for long periods of time.

To better understand the T2.2 strain removal ability, kinetic models were built in order to obtain the best fit for the experimental results (Figure 4a). When plotting both models, it was observed that the Weber–Morris model underestimated the points of high removal capacity and overestimated those of slow removal. On the other hand, the allometric model fitted well in the zone of high removal that occurred within the first 48 h, although it overestimated the zone of slow removal. In the last point, the allometric model was the one that best adjusted to the experimental data with a relative error of 2.36% while that of the Weber–Morris model was 3.88%. This behavior is corroborated with the results of the coefficient of determination $R^2$, which was 0.91 for the Weber–Morris model and 0.97 for the allometric model. Therefore, this last-mentioned model is the one that best fits the elimination behavior of the T2.2 strain.

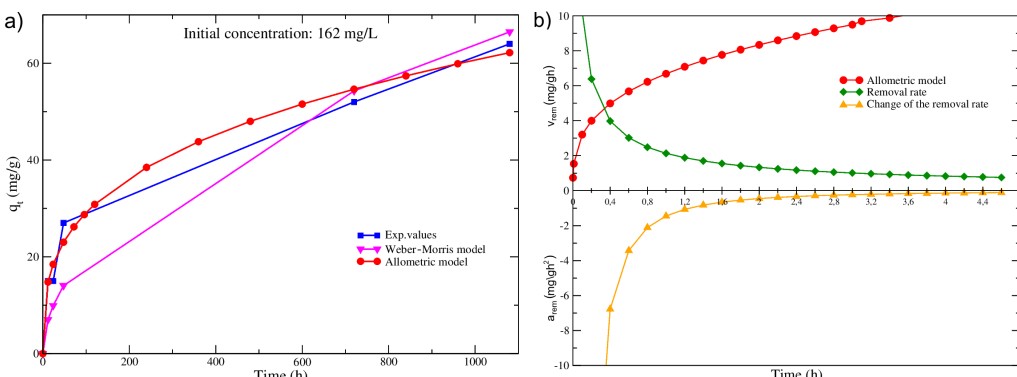

**Figure 4.** Experimental and kinetic models curves. (**a**) Hg removal capacities of studied models ($q_t$) at different times of the removal test. (**b**) Rate and acceleration of the removal capacity obtained by the allometric model.

Using the allometric model, removal rates and removal accelerations ($v_{rem}$, $a_{rem}$) were calculated to analyze the saturation behavior of strain T2.2 (Figure 4b). It can be seen that the rapid adsorption of mercury occurs within the first 4.6 h after starting the test, obtaining a maximum adsorption rate of 1.23 mg· $g^{-1}$· $h^{-1}$ with a deceleration of 0.11 mg· $g^{-1}$· $h^{-1}$. With this model, the saturation time of the T2.2 strain was calculated to occur at 1560 h (65 days), with a constant removal rate (0.014 mg· $g^{-1}$· $h^{-1}$). The use of allometric kinetic models has been widely used successfully in bioremediation studies of contaminated soils [79,80]. Liu et al. used these allometric models to understand the biodegradation processes of chemical compounds using bacterial strains of *Pseudomonas sp.* cbp1-3. [81]

## 4. Conclusions

The Hg-resistant bacterial strain *Zhihengliuella sp.* T2.2 was isolated from the Sacocha annex in Arequipa. Its morphology and biochemical characteristics were evaluated by examining the morphology of the colony, the Gram-staining technique, and the catalase and oxidase tests, which allowed the isolated strain to be identified as Gram-positive cocci, sensitive to levofloxacin. Furthermore, the 16S rRNA sequencing and its sequence analysis showed that they were new species found. An advanced analysis using the cold vapor method was carried out to determine the Hg removal efficiency. This strain was able to remove almost 40% $Hg^{2+}$ from the culture media with initial $Hg^{2+}$ concentrations of 162 mg·$L^{-1}$ in 1080 h. The kinetic model that best adjusted to the experimental data was the allometric model, which reproduced the behavior in the zone of high removal and adjusted to the zone of low removal. With this model, it was possible to determine the saturation point of the strain, which occurred after 65 days of contact, with a removal rate of 0.014 mg· $g^{-1}$· $h^{-1}$.

T2.2 bacterial isolates are promising for the remediation of Hg-contaminated water and soils. Its high resistance and long-term removal capacity of Hg make this variety an attractive option for use. Advanced in vitro and in vivo tests must be performed to fully assess the efficiency of bacteria in remediating heavy metal contamination in nature.

**Author Contributions:** Conceptualization, F.F.-F., B.G. and J.A.A.-P.; methodology, J.A.A.-P. and P.L.G.-B.; software, C.F.-M. and B.G.; validation, P.L.G.-B.; formal analysis, B.G. and C.F.-M.; investigation, P.L.G.-B. and J.A.A.-P.; resources, C.F.-M. and P.L.G.-B.; data curation, J.A.A.-P. and A.C.-H.; writing—original draft preparation, J.A.A.-P. and C.F.-M.; writing—review and editing, B.G.; visualization, J.C.B.-O. and A.C.-H.; supervision, B.G. and J.A.A.-P.; project administration, P.L.-C.; funding acquisition, P.L.-C. All authors have read and agreed to the published version of the manuscript.

**Funding:** The author thanks the financial support of the UCSM.

**Institutional Review Board Statement:** Not applicable.

**Informed Consent Statement:** Not applicable.

**Data Availability Statement:** Not applicable.

**Acknowledgments:** We are grateful for the financial support received through projects 24150-R-2017VRINV, 25175-R-2018VRINV and 23824-R-2016VRINV from UCSM internal funding for developed this research.

**Conflicts of Interest:** The authors declare no conflict of interest.

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
