# Peer review of "Identification and Characterization of Peruvian Native Bacterial Strains as Bioremediation of Hg-Polluted Water and Soils Due to Artisanal and Small-Scale Gold Mining in the Secocha Annex, Arequipa"

_sustainability, doi:10.3390/su14052669_

Round 1

Reviewer 1 Report

The authors met all the recommendations of the previous review, thereby substantially improving the quality of the article. The English has been carefully revised. 
Apart from the minor correction of placing the units in table 1, the article is now ready for publication.

Reviewer 2 Report

The manuscript has been substantially improved. However, some comments and suggestions can be found in the PDF file. It is not clear why the manuscript title includes water and scarce data about it is shown/discussed in the manuscript. In the conclusions section, the authors state that the findings in Hg removal can be helpful in the remediation of soils and water. Figure 1 definitely needs improvement; can the authors show the sampling point? Why is the settlement (satellite image) included in figure 1? It is not clear if sampling was carried out within the urban zone. 
